

# On the challenges of drawing conclusions from *p*-values just below 0.05

Daniël Lakens

School of Innovation Sciences, Eindhoven University of Technology, Eindhoven, The Netherlands

## ABSTRACT

In recent years, researchers have attempted to provide an indication of the prevalence of inflated Type 1 error rates by analyzing the distribution of *p*-values in the published literature. *De Winter & Dodou (2015)* analyzed the distribution (and its change over time) of a large number of *p*-values automatically extracted from abstracts in the scientific literature. They concluded there is a 'surge of *p*-values between 0.041–0.049 in recent decades' which 'suggests (but does not prove) questionable research practices have increased over the past 25 years.' I show the changes in the ratio of fractions of *p*-values between 0.041–0.049 over the years are better explained by assuming the average power has decreased over time. Furthermore, I propose that their observation that *p*-values just below 0.05 increase more strongly than *p*-values above 0.05 can be explained by an increase in publication bias (or the file drawer effect) over the years (cf. *Fanelli, 2012*; *Pautasso, 2010*, which has led to a relative decrease of 'marginally significant' *p*-values in abstracts in the literature (instead of an increase in *p*-values just below 0.05). I explain why researchers analyzing large numbers of *p*-values need to relate their assumptions to a model of *p*-value distributions that takes into account the average power of the performed studies, the ratio of true positives to false positives in the literature, the effects of publication bias, and the Type 1 error rate (and possible mechanisms through which it has inflated). Finally, I discuss why publication bias and underpowered studies might be a bigger problem for science than inflated Type 1 error rates, and explain the challenges when attempting to draw conclusions about inflated Type 1 error rates from a large heterogeneous set of *p*-values.

Corresponding author
Daniël Lakens, d.Lakens@tue.nl

## INTRODUCTION

In recent years, researchers have become more aware of how flexibility during the data-analysis can increase false positive results (e.g., *Simmons, Nelson & Simonsohn, 2011*). If the true Type 1 error rate is substantially inflated, for example because researchers analyze their data until a *p*-value smaller than 0.05 is observed, the robustness of scientific knowledge can substantially decrease. However, as *Stroebe & Strack* (*2014*, p. 60) have pointed out: '*Thus far, however, no solid data exist on the prevalence of such research practices.*' Some researchers have attempted to provide an indication of the prevalence of inflated Type 1 error rates by analyzing the distribution of *p*-values in the published

literature. The idea is that inflated Type 1 error rates lead to 'a peculiar prevalence of $p$-values just below 0.05' (*Masicampo & Lalande, 2012*), the observation that "just significant" results are on the rise' (*Leggett et al., 2013*), and that '$p$-hacking is widespread throughout science' (*Head et al., 2015*).

Despite the attention grabbing statements in these publications, the strong conclusions these researchers have drawn do not follow from the empirical data. The pattern of a peak of $p$-values just below $p = 0.05$ observed by *Leggett et al. (2013)* does not replicate in other datasets of $p$-value distributions for the same journal in later years (*Masicampo & Lalande, 2012*), in psychology in general (*Hartgerink et al., unpublished data*; *Kühberger, Fritz & Scherndl, 2014*), or in scientific journals in general (*De Winter & Dodou, 2015*). The peak in $p$-values observed in *Masicampo & Lalande (2012)* is only surprising compared to an incorrectly modeled $p$-value distribution that ignores publication bias and its effect on the frequency of $p$-values above 0.05 (*Lakens, 2014a*, see also *Vermeulen et al., in press*). The 'widespread' $p$-hacking observed by Head and colleagues (*2015*) disappears after controlling for a simple confound (*Hartgerink, 2015*).

Recently, *De Winter & Dodou (2015)* have contributed to this emerging literature on $p$-value distributions and concluded that there is a 'surge of $p$-values between 0.041–0.049 in recent decades'. They improved upon earlier approaches to analyze $p$-value distributions by comparing the percentage of $p$-values over time (from 1990–2013). Two observations in the data they collected could seduce researchers to draw conclusions about a rise of $p$-values just below a significance level of 0.05. The first observation the authors report is how from 1990 to 2013 $p$-values between 0.041 and 0.049 rose more strongly than the percentage of $p$-values between 0.051–0.059. The second observation is that the percentage of $p$-values that falls between 0.041–0.049 has increased more than the increase in the percentage of $p$-values between 0.001–0.009, 0.011–0.019, 0.021–0.029, and 0.031–0.039 from 1990 to 2013 [1]. The authors (2015, p. 37) conclude that: "The fact that $p$-values just below 0.05 exhibited the fastest increase among all $p$-value ranges we searched for suggests (but does not prove) that questionable research practices have increased over the past 25 years."

I will explain why the data does not suggest an increase in 'questionable research practices'. First, I will discuss how the relatively stronger increase in $p$-values just below $p = 0.05$ compared to $p$-values just above $p = 0.05$ is not caused by a change over time in the percentage of $p$-values just *below* 0.05, but by a change over time in the percentage of $p$-values *above* 0.05. Perhaps surprisingly, $p$-values just above 0.05 increase much less than all other $p$-values. This might be due to a stricter interpretation of $p < 0.05$ as support of a hypothesis, and less leniency for 'marginally significant' $p$-values just above this threshold. Second, I will explain why the relatively high increase in $p$-values between 0.041–0.049 over the years can easily be accounted for by a decrease in the average power of studies. At the same time, I will illustrate why this increase in $p$-values just below 0.05 is unlikely to emerge due to an inflation of the Type 1 error rate due to optional stopping or trying out multiple analyses until $p < 0.05$. I want to explicitly note that it was possible to provide these alternative interpretations of the data because *De Winter & Dodou (2015)* shared all data and analysis scripts online. While I criticize their interpretation of data, I applaud their

[1] The authors also analyze $p$-values with 2 digits (e.g., $p = 0.04$), which reveal similar patterns, but here I focus on the three digit data, which focuses on $p$-values between (for example) 0.041–0.049 because trailing zeroes (e.g., $p = 0.040$) are rarely reported.

adherence to open science principles, which greatly facilitated cumulative science. Most importantly, the main point of this article is to highlight the challenges in drawing conclusions about inflated Type 1 error rates based solely on a large heterogeneous set of $p$-values.

As I have discussed before (*Lakens, 2014a*), it is essential to use a model of $p$-value distributions before drawing conclusions about the underlying reasons for specific distributions of $p$-values extracted from the scientific literature. A model of $p$-value distributions consists of four different factors. First, the $p$-value distribution depends on the number of studies where the null-hypothesis (H0) is true, and the number of studies where the alternative hypothesis (H1) is true. Second, the $p$-values for studies where H1 is true depend upon the power of the studies. Statistical power is the probability that a study yields a statistically significant effect, if there is a true effect to be found. Power is determined by the significance level, the sample size, and the effect size. Third, $p$-values for studies where H0 is true depend upon the Type 1 error rate chosen by the researcher (e.g., 0.05), and any possible mechanisms through which the Type 1 error rate is inflated beyond the nominal Type 1 error rate set by the researcher. When I talk about inflated Type 1 error rates in this article, I explicitly mean flexibility in dependent tests that are performed on the data (e.g., by performing a test after every few participants, flexibly deciding to exclude participants, or dropping or combining measurements) that have the goal to lead to a statistically significant result. When these statistical tests are dependent (e.g., analyzing the data after 20 participants, and analyzing the same data again after adding 10 additional participants) the Type 1 error rate inflation has a specific pattern where $p$-values between 0.041–0.049 become somewhat more likely than smaller $p$-values.

And finally, the $p$-value distribution in the published literature is influenced by publication bias. Publication bias is the tendency to publish statistically significant results (both because authors are more likely to submit those articles, as that editors and reviewers are more likely to evaluate such manuscripts more positively). The threshold at which $p$-values indicate a statistically significant result, as well as the leniency towards 'marginally significant' findings, both influence the frequency of observed $p$-values in the literature. It is important to look beyond simplistic comparisons between $p$-values just below 0.05 and $p$-values in other parts of the $p$-value distribution if the observed $p$-values are not explicitly related to a model consisting of the four factors that determine $p$-value distributions.

## ARE *P*-VALUES BELOW 0.05 INCREASING, OR *P*-VALUES ABOVE 0.05 DECREASING?

*De Winter & Dodou (2015)* show there is a relatively stronger increase over time in $p$-values between 0.041–0.049 than in $p$-values between 0.051–0.059 (see for example their Fig. 9). The data is clear, but the reason for this difference is not, and it is not explored by the authors. Although all $p$-values are increasing over time, the real question is whether $p$-values below $p = 0.05$ are increasing more, or $p$-values above $p = 0.05$ are increasing less. A direct comparison is difficult, because a comparison across the $p = 0.05$ boundary is influenced by publication bias. If publication bias increases, and less non-significant results end up in the published literature due to the file-drawer problem, the percentage of papers

reporting $p$-values smaller than 0.05 will also increase (even when there is no increase in $p$-hacking). Indeed, both *Pautasso (2010)* as *Fanelli (2012)* have provided support for the idea that negative results have been disappearing from the literature, which raises the possibility that the relative differences in $p$-values between 0.041–0.049 and 0.051–0.059 observed by *De Winter & Dodou (2015)* are actually caused by a relative decrease in $p$-values between 0.051–0.059.

By comparing the relative differences between $p$-values between 0.031–0.039 and 0.041–0.049 over the years on the one hand, and 0.051–0.059 and 0.061–0.069 on the other hand, we can examine whether there is an increase in $p$-values between 0.041–0.049 (due to an increase in the Type 1 error rate), or an increase in publication bias (or the file-drawer problem), which leads to a lower percentage of $p$-values between 0.051–0.059. If there is an increase in the Type 1 error rate due to flexibility in the data analysis, the biggest differences over time should be observed just below $p = 0.05$ (in line with the idea of a surge of $p$-values between 0.041–0.049). However, there are reasons to assume the biggest difference will be observed in $p$-values just above $p = 0.05$. As *Lakens (2014a)* noted, there seems to be some tolerance for $p$-values just above 0.05 to be published, as indicated by a higher prevalence of $p$-values between 0.051–0.059 than would be expected based on the power of statistical tests and an equal reduction of all $p$-values above 0.05 due to the file-drawer problem. If publication bias becomes more severe, we might expect a reduction in the tolerance for 'marginally significant' $p$-values just above 0.05, and the largest changes in ratios should be observed *above $p = 0.05$*.

Across the three time periods (1990–1997, 1998–2005, and 2006–2013) the ratio of $p$-values between 0.031–0.039 to $p$-values between 0.041–0.049 is pretty stable: 1.13, 1.09, and 1.11, respectively. The ratio of $p$-values between 0.051–0.059 to $p$-values between 0.061–0.069 shows a surprisingly large reduction over the years: 2.27, 1.94, and 1.79, respectively. It is important to note that flexibly analyzing data with the goal to be able to report a significant finding leads to a change in the $p$-value distribution both above as below $p = 0.05$. However, the ratio of $p$-values between 0.031–0.039 to $p$-values between 0.041–0.049 should change much more than the ratio of $p$-values between 0.051–0.059 to $p$-values between 0.061–0.069, because $p$-values are drawn from a relatively larger range above $p = 0.05$, to a relatively small range just below $p = 0.05$. This surprisingly large change in ratios over time for $p$-values 0.051–0.059 to 0.061–0.069 indicates that instead of an increase in the Type 1 error rate of $p$-values below 0.05, the real change over time happens in the $p$-values between 0.051–0.059.

The change over time in $p$-values just above $p = 0.05$ might be explained by an increasingly strong effect of the file-drawer problem. Where $p$-values between 0.051–0.059 (or 'marginally significant' results) might have been more readily accepted as support for the alternative hypothesis in 1990–1997, $p$-values just above 0.05 might no longer deemed strong enough support for the alternative hypothesis in 2005-2013. This idea is speculative, but seems plausible given the increase in publication bias over the years (*Fanelli, 2012*; *Pautasso, 2010*), which suggests that non-significant results are less likely to be published in recent years. It should be noted that $p$-values just above the 0.05 level are *still* more frequent

than can be explained just by the average power of the tests combined with publication bias that is equal for all $p$-values above 0.05 (cf. *Lakens, 2014a*). In other words, this data is in line with the idea that publication bias is still slightly less severe for $p$-values just above 0.05, even though this benefit of $p$-values just above 0.05 has become smaller over the years.

## HOW A CHANGES IN AVERAGE POWER OVER THE YEARS AFFECTS RATIOS OF *P*-VALUES BELOW 0.05

The first part of the title of the article by *De Winter & Dodou (2015)*, "A surge of $p$-values between 0.041–0.049" is based on the observation that the ratio of $p$-values between 0.041–0.049 increases more than the ratio of $p$-values between 0.031–0.039, 0.021–0.029, and 0.011–0.019. There are no statistics reported to indicate whether these differences in ratios are actually statistically significant, nor are effect sizes reported to indicate whether the differences are practically significant (or justify the term 'surge'), but the ratios do increase as you move from bins of low $p$-values between 0.001–0.009 to bins of high $p$-values between 0.041–0.049.

The first thing to understand is why none of the observed ratios are close to 1. The reason is that there is a massive increase in the percentage of abstracts of papers in which $p$-values are reported over the years. As *De Winter & Dodou* (*2015*, p. 15) note: "*In 1990, 0.019% of papers (106 out of 563,023 papers) reported a p-value between 0.051 and 0.059. This increased 3.6-fold to 0.067% (1,549 out of 2,317,062 papers) in 2013. Positive results increased 10.3-fold in the same period: from 0.030% (171 out of 563,023 papers) in 1990 to 0.314% (7,266 out of 2,317,062 papers) in 2013.*" *De Winter & Dodou (2015)* show $p$-values are finding their way into more and more abstracts, which points to a possible increase in the overreliance on null-hypothesis testing in empirical articles. This is an important contribution to the literature.

The main question is how these differences in the ratios across the 5 bins below $p = 0.05$ can be explained. *De Winter & Dodou (2015)* do not attempt to model their hypothesized mechanism by choosing values for the four factors of the model (the ratio of studies where H0 or H1 is true, the power of studies, the Type 1 error rate, and the presence of the file-drawer problem). However, this model contains all the factors that together completely determine the $p$-value distribution (except perhaps erroneously calculated $p$-values, which is also common, see *Hartgerink et al., unpublished data*; *Vermeulen et al., in press*). Therefore, the hypothesis that flexibility in the data-analysis increases the Type 1 error rate must be translated into specific parameters for the factors in this model. It is only possible to explain the relative differences between the ratios of the different bins of $p$-values if we allow at least one of the parameters of the model to change over time. Because we are focusing on the $p$-values below 0.05 we can ignore the file drawer problem, assuming all disciplines that report $p$-values in abstracts use $\alpha = 0.05$ (this is not true, but we can assume it applies to the majority of articles that are analyzed). The three remaining possibilities are a change in the average power of studies over time, a change in the inflated Type 1 error rate over time, and a change in the ratio of studies where H0 or H1 is true. I will discuss each of these three possible explanations in turn.

**Table 1 Expected percentage of *p*-values between 0.001–0.049 based on 42% and 55% power.** Note that columns do not sum to 0.55 and 0.42 because some *p*-values are not included in the analysis (e.g., *p*-values between 0.049–0.050).

| | Expected *p*-values per bin with 55% power | Expected *p*-values per bin with 42% power |
|---|---|---|
| *p*0.001–*p*0.009 | 0.300 | 0.199 |
| *p*0.011–*p*0.019 | 0.085 | 0.072 |
| *p*0.021–*p*0.029 | 0.056 | 0.051 |
| *p*0.031–*p*0.039 | 0.042 | 0.034 |
| *p*0.041–*p*0.049 | 0.034 | 0.033 |

## Changes in power over time

We can relatively easily reconstruct the observed data purely based on differences in the average power across the years. Remember that the distribution of *p*-values depends on the statistical power of the studies (or the average power of multiple subsets of studies, if heterogeneity in power is substantial), which is itself a function of the true effect size, the significance level, and the sample size (for formulas, see *Cumming, 2008*; *Lakens & Evers, 2014*, for an online app, see http://rpsychologist.com/d3/pdist/). It is not difficult to model the ratios observed by *De Winter & Dodou (2015)* under the assumption that power decreases from 1990 to 2013. For example, if we assume the average power of studies was 55% in 1990, and 42% in 2013, we will (given a large enough sample) observe the *p*-value distribution across the 5 bins as detailed in Table 1, with 29.86% of the *p*-values falling between 0.001 and 0.009 in 1990, but only 19.93% of *p*-values falling between 0.001 and 0.009 in 2013. This is just the *p*-value distribution as a function of the power of the tests.

The total number of studies analyzed by *De Winter & Dodou (2015)* was 561,516 in 1990, and 2,311,772 in 2013. Because the authors note how the percentage of statistically significant *p*-values reported in abstracts has increased by 10% over the years, and I chose 0.01% in 1990 and 0.1% in 2013 as the percentage of abstracts that report *p*-values (column 1 and 2 in Table 2). Assuming the average power was 55% in 1990 and 42% in 2013, we can calculate the expected number of observed *p*-values in 1990 and 2013 by simply multiplying the total number of articles (e.g., 561,516) by the percentage of articles reporting *p*-values (e.g., 0.01), multiplied by the percentage of *p*-values expected in each *p*-value bin based on the assumed power (e.g., 0.300). The number of reconstructed *p*-values is presented in Table 2, columns 3 and 4. These numbers closely resemble the absolute number of *p*-values observed by *De Winter & Dodou* (*2015*, Table 2, columns 5 and 6), indicating the chosen parameters for the model can reproduce the observed data.

Following *De Winter & Dodou (2015)*, the fraction of the observed *p*-values in each of the five *p*-value bins can now be calculated by dividing the number (*N*) of *p*-values in a specific bin for a specific year (e.g., *N* = 1,676) by the total (*T*) number of *p*-values (e.g., *T* = 561,516) to get the fractions for 1990 and 2015 (e.g., 1,676/561,516 × 100 = 0.299). The reconstructed fractions (Table 3, column 1 and 2) are very similar to the observed fractions by De Winter and Dodou (columns 4 and 5). The main dependent

**Table 2** Percentage of papers that report *p*-values in abstracts, and the number of reconstructed and observed (*De Winter & Dodou, 2015*) *p*-values between 0.001–0.049 in 1990 and 2013 for each bin.

| | % *p*-values in abstracts 1990 | % *p*-values in abstracts 2013 | Reconstructed # *p*-values 1990 | Reconstructed # *p*-values 2013 | Observed # *p*-values 1990 | Observed # *p*-values 2013 |
|---|---|---|---|---|---|---|
| *p*0.001–*p*0.009 | 0.01 | 0.1 | 1,676 | 46,064 | 1,770 | 44,970 |
| *p*0.011–*p*0.019 | 0.01 | 0.1 | 480 | 16,690 | 462 | 14,885 |
| *p*0.021–*p*0.029 | 0.01 | 0.1 | 315 | 11,698 | 268 | 10,630 |
| *p*0.031–*p*0.039 | 0.01 | 0.1 | 237 | 9,189 | 240 | 9,108 |
| *p*0.041–*p*0.049 | 0.01 | 0.1 | 190 | 7,629 | 178 | 8,250 |

**Table 3** Ratio of fractions of reconstructed *p*-values and *p*-value ratios observed by *De Winter & Dodou (2015)* between 0.001–0.049 for 1990 and 2013.

| *p*-value bin | Reconstructed fraction $N/T$ 1990 | Reconstructed fraction $N/T$ 2013 | Reconstructed 1990/2013 ratio of fractions | Observed fraction $N/T$ 1990 | Observed fraction $N/T$ 2013 | Observed 1990/2013 ratio of fraction |
|---|---|---|---|---|---|---|
| 0.001–0.009 | 0.299 | 1.993 | 6.674 | 0.315 | 1.945 | 6.17 |
| 0.011–0.019 | 0.085 | 0.722 | 8.454 | 0.082 | 0.644 | 7.83 |
| 0.021–0.029 | 0.056 | 0.506 | 9.017 | 0.048 | 0.460 | 9.63 |
| 0.031–0.039 | 0.042 | 0.398 | 9.417 | 0.043 | 0.394 | 9.21 |
| 0.041–0.049 | 0.034 | 0.330 | 9.740 | 0.032 | 0.367 | 11.28 |

variable De Winter and Dodou analyze is the fraction in 1990 divided by the fraction in 2013 (e.g., 0.299/1.993 = 6.674, basically a ratio of fractions), and we can see the reconstructed ratios of fractions of 1990/2013 (column 3) closely resemble the observed ratios of fractions (column 6).

The reconstruction is close, but not perfect, for a number of reasons. First of all, there are very few data points from 1990, which will lead to substantial variation between expected and observed frequencies. For example, the reconstructed 1990/2013 ratio of fractions in the 0.021–0.029 bin (9.017) is smaller than the reconstructed ratio of fractions in the 0.031–0.039 bin (9.417), but the pattern is reversed in the observed data (9.63 and 9.21, respectively). However, if we calculate the same ratios of fractions for 2013 with all other preceding years (e.g., 1991, 1992, 1993, etc.) we find a smaller ratio of fractions in the 0.021–0.029 bin than in the 0.031–0.039 bin for *all* the remaining 21 comparisons (see Table 4, columns 3 and 4). In other words, the model correctly predicts the ratios of fractions in 21 out of 22 comparisons between years, even when we chose the parameters for the model based on the 1990/2013 comparison. This provides strong support for the validity of the model that is used to reconstruct the ratios.

The model based on power differences similarly predicts that ratios for *p*-values between 0.031–0.039 should be very similar to those between 0.041–0.049. The predicted 1990/2013 ratio of fractions in the 0.031–0.039 bin is 9.417, and the predicted ratio of fractions for the 0.041–0.049 bin is 9.740, while the difference in observed ratios of fractions is much larger (9.21 and 11.28, respectively, see column 6 in Table 3). If the true difference is large (following the observed ratios) the ratio of fractions in the 0.041–0.049 bin should be

**Table 4** Ratios of fractions of the percentage of *p*-values in abstracts for 2013 relative to the percentages in 23 preceding years, for the 5 *p*-value bins.

| Year compared to 2013 | Year/2013 ratio of fractions per *p*-value bin | | | | |
|---|---|---|---|---|---|
| | 0.001–0.009 | 0.011–0.019 | 0.021–0.029 | 0.031–0.039 | 0.041–0.049 |
| 1990 | 6.17 | 7.83 | 9.63 | 9.22 | 11.26 |
| 1991 | 5.10 | 6.81 | 6.90 | 8.02 | 9.19 |
| 1992 | 4.07 | 5.04 | 5.72 | 6.30 | 6.55 |
| 1993 | 3.03 | 4.06 | 4.39 | 5.01 | 4.92 |
| 1994 | 2.56 | 3.26 | 3.61 | 4.28 | 4.31 |
| 1995 | 2.10 | 2.72 | 2.98 | 3.22 | 3.36 |
| 1996 | 1.62 | 2.23 | 2.26 | 2.48 | 2.52 |
| 1997 | 1.42 | 1.84 | 2.15 | 2.19 | 2.16 |
| 1998 | 1.26 | 1.76 | 1.85 | 2.15 | 1.95 |
| 1999 | 1.12 | 1.52 | 1.64 | 1.70 | 1.70 |
| 2000 | 1.02 | 1.29 | 1.39 | 1.42 | 1.50 |
| 2001 | 0.95 | 1.19 | 1.27 | 1.34 | 1.25 |
| 2002 | 0.88 | 1.08 | 1.09 | 1.17 | 1.15 |
| 2003 | 0.77 | 0.88 | 0.90 | 0.97 | 0.98 |
| 2004 | 0.67 | 0.76 | 0.78 | 0.81 | 0.81 |
| 2005 | 0.57 | 0.64 | 0.66 | 0.67 | 0.67 |
| 2006 | 0.52 | 0.57 | 0.59 | 0.60 | 0.62 |
| 2007 | 0.47 | 0.51 | 0.51 | 0.54 | 0.54 |
| 2008 | 0.42 | 0.45 | 0.46 | 0.45 | 0.48 |
| 2009 | 0.39 | 0.42 | 0.41 | 0.43 | 0.42 |
| 2010 | 0.36 | 0.38 | 0.38 | 0.40 | 0.39 |
| 2011 | 0.31 | 0.33 | 0.32 | 0.33 | 0.33 |
| 2012 | 0.27 | 0.28 | 0.27 | 0.27 | 0.28 |

consistently higher than the 0.031–0.039 bin across all years. If the reconstructed ratios are true, the difference between the two bins should be less pronounced across all year. When comparing 2013 to each of the 23 preceding years, the ratio of fractions in the 0.041–0.049 bin (see Table 4) is higher than for *p*-values in the 0.031–0.039 bin in only 12 out of 23 comparisons (52% of the time). This can hardly be called a 'surge' of *p*-values between 0.041–0.049. This observation is not in line with the idea that the Type 1 error rate has increased, because an increase in Type 1 error rates due to flexibility in the data analysis typically assumes *p*-values between 0.041–0.049 increase more strongly than *p*-values between 0.031–0.039 (e.g., *Head et al., 2015*; *Leggett et al., 2013*; *Masicampo & Lalande, 2012*).

Obviously a model that explains the observed *p*-value distribution only based on a change in the average power of the studies (and sets the other factors to zero) is not likely to reflect the true state of affairs in the real world. Although we lack data about changes in the ratio of true to false effects examined over time (a worthwhile research question in itself), it seems reasonable to at least entertain the possibility of some changes over time for this factor of the model. For now, the most important conclusion is that a change in power over time can mathematically account for the observed changes in ratios of fractions in the

different $p$-value bins. Moreover, the idea that power decreases over time is theoretically plausible, since such a decrease in power over time has been observed in some disciplines, such as psychology (*Sedlmeier & Gigerenzer, 1989*), and the values of the parameters (55% and 42% power) are plausible. At the same time, we can be certain power varies substantially across studies and research disciplines (e.g., *Button et al., 2013*; *Sedlmeier & Gigerenzer, 1989*), and therefore the $p$-value distribution can be more accurately modeled by summing multiple $p$-value distributions across different research areas.

## Changes in Type 1 error rates over time

Let's assume the average power has not changed over time, and instead try to reconstruct the observed data by *De Winter & Dodou (2015)* based on a change in the Type 1 error rates over time. The assumption is that there should be an increase in $p$-values just below $p = 0.05$ because questionable research practices increase the number of false positives (*De Winter & Dodou*, p. 6). A false positive or Type 1 error occurs when the null-hypothesis is true, but a statistically significant result is observed. When Type 1 error rates are not inflated (or *nominal*) 5% of the studies will observe $p$-values smaller than 0.05. De Winter and Dodou focus on $p$-values between 0.041–0.049 in the analysis of $p$-values with three digits (where $p$-values such as 0.040 will be absent because researchers are most likely to write 0.04 instead of 0.040), which means 0.8% of the time (0.049 minus 0.041) a $p$-value within each of the five $p$-value bins will be observed[2]. By increasing the Type 1 error rate above 0.8%, and doing so more strongly for higher $p$-value bins, we can attempt to reconstruct the consequences of an increase in false positives in the literature on the observed ratios of $p$-values over time.

The observed ratios of fractions by *De Winter & Dodou (2015)* show the 1990/2013 ratio is the smallest for $p$-values between 0.001–0.009 (i.e., 6.17), and substantially higher for $p$-values between 0.011 and 0.049 (see Table 3, last column). It is important to realize that the 1990/2013 ratios (see row 1, Table 4) with the large difference between the 0.031–0.039 and 0.041–0.049 bins (9.21 and 11.28, respectively) is an outlier—the ratios in the two $p$-value bins are on average the same between 1992 and 2012. The pattern in the ratios for the 1990/2013 ratios can be reproduced based on inflated Type 1 error rates. However, reconstructing the ratios for the last twenty years requires an inflated Type 1 error rate that is unlikely to occur in real life.

One attempt to specify the parameters to model the ratios (but not the absolute values, because this proved to be too difficult solely based on an inflated Type 1 error rate) is presented in Table 5 (column 5). The ratio of studies where H0 is true to studies where H1 is true is set to 1 (a hypothesis is equally likely to be true or false), and the average power is assumed to be 57.5%. The Type 1 error rate inflation over time has been increased based on a modest $p$-hacking strategy (collecting 50 participants in each condition, analyzing the data after every 10 participants until 100 have been collected). Although this would be substantial (it assumes $p$-hacking occurs in *all* studies where H0 is true), it is not impossible.

However, such a pattern of Type 1 error rates does not predict the ratio of fractions in the last 20 years, which were on average equal for the 0.031–0.039 and 0.041–0.049 bins.

[2] Because researchers round numbers to the nearest 3 digits, one might argue that instead of 0.8% of $p$-values between 0.041–0.049 when the null hypothesis is true, *De Winter & Dodou (2015)* actually focus on $p$-values between 0.0405–0.0495, which would mean 0.9% of the time a $p$-value within a single $p$-value bin will be observed. Using 0.8% or 0.9% Type 1 error rates in each bin does not influence the reconstruction purely based on changes in power over time, and does not have consequences for any of the conclusions based on the reconstructions.

**Table 5** Type 1 error rates, absolute number of reconstructed Type 1 errors between 0.001–0.049 from 1990 to 2013, and their ratio.

| | Type 1 error rate 1990 | Type 1 error rate 2013 | Significant *p*-values 1990 | Significant *p*-values 2013 | Reconstructed 1990/2013 ratio of fractions |
|---|---|---|---|---|---|
| *p*0.001–*p*0.009 | 0.008 | 0.009 | 1,899 | 50,338 | 6.44 |
| *p*0.011–*p*0.019 | 0.008 | 0.014 | 581 | 17,072 | 7.14 |
| *p*0.021–*p*0.029 | 0.008 | 0.018 | 408 | 13,706 | 8.16 |
| *p*0.031–*p*0.039 | 0.008 | 0.022 | 327 | 12,755 | 9.47 |
| *p*0.041–*p*0.049 | 0.008 | 0.028 | 279 | 13,262 | 11.56 |

If we would change the Type 1 error rates to reconstruct the very similar ratios in the majority of the years in the 0.031–0.039 and 0.041–0.049 bins only based on a change in Type 1 error rates, the Type 1 error rates should be equal across these bins, or even slightly lower in the 0.041–0.049 bin compared to the 0.031–0.039 bin. This would be opposite to predictions based on flexibility during the data analysis (*De Winter & Dodou, 2015*; *Head et al., 2015*; *Leggett et al., 2013*; *Masicampo & Lalande, 2012*). The *p*-value distribution for true effects (based on the power of the studies) leads to a lower frequency of *p*-values in the 0.041–0.049 bin than in the 0.031–0.039 bin. A uniform inflation of the Type 1 error rate (e.g., increasing Type 1 error rates from 0.008 to 0.02 in all *p*-value bins) would add the same number of false positive *p*-values to each bin, but because the frequency of true *p*-values decreases from bins with low *p*-values to bins with high *p*-values, the relative increase is stronger in the 0.041–0.049 bin than in the 0.031–0.039 bin because the extra *p*-values from false positives constitute a relatively larger increase. As Table 4 shows the ratios in the 0.031–0.039 bins and 0.041–0.049 bin are very similar in most of the years. Therefore, reconstructing these ratios would actually require a lower increase in the Type 1 error rate in the 0.041–0.049 bin than in the 0.031–0.039 bin.

Researchers can (and probably do) *p*-hack studies where H1 is true. If such behavior increases over time, we can expect the percentage of *p*-values from true effects in the 0.041–0.049 bin to increase in 2013 compared to 1990. If we would incorporate this effect in the model, the relative increase over time in the 0.041–0.049 bin would be even stronger compared to the 0.031–0.039 bin. Again, we do not see huge differences in the ratios between the 0.031–0.039 and 0.041–0.049 bins, which makes the explanation based on flexibility in the data analysis even less likely.

## Changes in the ratio of true to false effects that are examined

The third factor that could influence the ratios is the percentage of studies where true effects are examined. A hypothesis is either true or not. The current analysis focuses on statistically significant findings in the published literature (i.e., with *p*-values smaller than 0.05). This means that *p*-values are either true positives (when H1 is true) or false positives (when H0 is true). The ratios calculated by *De Winter & Dodou (2015)* can change over time based on the idea that for each published study in 2013 that examined a true hypothesis, there is a much greater percentage of studies that examined a false hypothesis than in 1990. Most of the studies that examined a false hypothesis will end-up in the file

drawer, but we can assume (most of) the false positives end up in the literature. This greater number of false positives among the true positives in 2013 compared to 1990 could, theoretically, explain the change in ratios over time.

However, the effect of a change over time of the ratio of true positives to false positives on the *p*-value ratios in each of the five bins is really small. To account for the observed ratios by *De Winter & Dodou (2015)*, the changes in these ratios need to be quite substantial for a close reconstruction of the observed data. For example, when keeping the power (i.e., 55% power) and Type 1 error rate (i.e., 0.08% in each *p*-value bin) constant over time, the observed ratios of fractions (although again, not the observed absolute number of *p*-values) can be reconstructed purely based on a change in the H0/H1 ratio over time if we assume it was 10 times more likely to examine a true hypothesis compared to a false hypothesis in 1990, and at the same time assume it is 10 times more likely to examine a false hypothesis compared to a true hypothesis in 2013 (for details, see Supplemental Information). That is a truly massive change over time, where scientists are 100 times more likely to examine a false hypothesis than a true hypothesis in 2013 compared to in 1990. Without empirical data, it is not possible to conclusively reject this alterative explanation, but it seems highly implausible. More subtle changes over time might exist, and it is worthwhile to get data on the ratio of true to false hypotheses that researchers examine. However, with respect to the current question, it is unlikely this factor of the model underlies the ratios observed by *De Winter & Dodou (2015)*.

To summarize, we can easily reconstruct the observed ratios by assuming a relatively small decrease in power over the years (e.g., from 55% to 42%). On the other hand, while increases in Type 1 error rates can be used to reconstruct the observed ratios, the pattern of inflated Type 1 errors across the 5 bins of *p*-values is unlikely to emerge in real life, and the required difference in the ratio of true to false hypotheses researchers examine is even less likely. Therefore, I conclude it is not very likely to be true that there is a 'surge of *p*-values between 0.041–0.049', nor that these data suggest there is an increase in questionable research practices over the last 25 years. A more plausible explanation is a small reduction in the average power of experiments over the last twenty years. This could be caused by a stronger focus on smaller effects over the years, after many larger effects have already been uncovered and researchers focus more on moderators of known effects, or smaller novel effects.

## DISCUSSION

I hope to have illustrated the difficulty in accounting for the observed data by *De Winter & Dodou (2015)* purely based on an increase in inflated Type 1 error rates over the years. The search for evidence of an increase in questionable research practices in science in general, or at least across a large number of studies, is starting to mirror the search for the ether. After repeatedly claiming to observe a rise in *p*-values just below 0.05 without providing substantial evidence for such a rise (*De Winter & Dodou, 2015*; *Head et al., 2015*; *Leggett et al., 2013*; *Masicampo & Lalande, 2012*), and conflicting results across studies examining

this question, it is time that researchers investigating inflated Type 1 errors use a model of *p*-value distributions to check their assumptions.

Any criticisms on the suggestion that changes in power over time are a more likely explanation of the observed ratios than inflated Type 1 error rates should propose different parameters for the model of *p*-value distributions. Regardless of any disagreements about the specific values for the parameters, the model of *p*-value distributions I have used here, based on the ratio of true positives to false positives, the power of studies, the Type 1 error rate, and the effect of publication bias, is a mathematical reality. If researchers aim to draw conclusions about the Type 1 error rate in the literature based on *p*-value distributions, they need to specify values for the parameters in this model, and interpret how plausible these values are. My explanation for the observed *p*-value distribution in *De Winter & Dodou (2015)* contains clearly testable predictions, such as the leniency of reviewers and editors to accept marginally significant *p*-values as support for a hypothesis, and the prediction that, on average, power has decreased from 1990–2013. Testing these predictions in future studies allows the values I have chosen for the parameters in the model to be either falsified or corroborated.

Analyzing huge numbers of *p*-values, which come from studies with large heterogeneity in effect sizes, might lack the sensitivity to provide support for an inflation of Type 1 error rates. Furthermore, automatically retrieving *p*-values from abstracts does not allow researchers to identify all theoretically relevant tests that are performed in an article that might contain multiple studies. Articles also contain many statistical tests not related to the main hypothesis, and might describe non-significant results without reporting statistics (e.g., no other differences reached statistical significance). A better approach seems to be to perform targeted analyses of small sets of similar studies, which might be able to yield support for *p*-hacking (e.g., *Lakens, 2014b*; *Simonsohn, Nelson & Simmons, 2014*; *Van Assen, Van Aert & Wicherts, in press*). I do not doubt that Type 1 error rates are inflated in some lines of research, but the inflation and the percentage of experiments that examine a hypothesis where the null hypothesis is true need to be large to observe an effect on the *p*-value distribution across a large number of studies.

## CONCLUSIONS

Although it is important to control Type 1 error rates when performing statistical tests (e.g., *Lakens, 2014c*), I believe statistical power and publication bias due to the file drawer problem, and not *p*-hacking, are the biggest problems in the scientific literature. Without publication bias, Type 1 errors would be quite easily identified as they lead to follow-up research that will not observe the hypothesized effect. Even with publication bias, meta-analyses can identify sets of studies that lack evidential value, indicating the studies are a result of selection and reporting biases (*Lakens, Hilgard & Staaks, in press*; *Simonsohn, Nelson & Simmons, 2014*; *Van Assen, Van Aert & Wicherts, in press*). As the current analyses reveal, the *p*-value distributions in 1990 and 2013 in the data by *De Winter & Dodou (2015)* in 1990 and 2013 could be reproduced by assuming the average power of the studies was around 50%. This suggests that from the outset only half of the performed studies could

be expected to observe a statistically significant effect when the alternative hypothesis is true. This is clearly a huge waste of resources (especially in combination with publication bias). Inflated Type 1 errors and practices such as *p*-hacking have been very salient in recent years, but I believe it is worthwhile to point out that designing well-powered studies with high informational value (e.g., *Lakens & Evers, 2014*) and preventing publication bias (e.g., *Nosek & Lakens, 2014*) are at least, and I personally believe even more, important to improve our science.

## ACKNOWLEDGEMENTS

I want to thank De Winter and Dodou for sharing their data, assisting in the re-analysis, and reading an earlier version of this draft.

### Funding

The author declares there was no funding for this work.

### Competing Interests

The author declares there are no competing interests.

### Author Contributions

- Daniël Lakens conceived and designed the experiments, performed the experiments, analyzed the data, contributed reagents/materials/analysis tools, wrote the paper, prepared figures and/or tables, reviewed drafts of the paper.

### Data Deposition

The following information was supplied regarding the deposition of related data:
   Data is available from the Open Science Framework: https://osf.io/ms4x6/.

### Supplemental Information

Supplemental information for this article can be found online at http://dx.doi.org/10.7717/peerj.1142#supplemental-information.

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
