# Peer review of "On the challenges of drawing conclusions from *p*-values just below 0.05"

_PeerJ, doi:10.7717/peerj.1142_

## Round 0.1 · original submission · Major Revisions

· Academic Editor

Major Revisions

I welcome the reanalysis that you performed and hope you are willing to revise to make it an even better paper. In the revision, please take note of the comments of the reviews. From my own reading, I have the following comments:

The title is ‘grabbing’ but I find the word ‘incorrect conclusions’ too strong and not defendable from the current ms. I suggest either to put the best fitting alternative model/explanation in the title, or something as “Alternative explanations for….” Position the paper less as a commentary on De winter and Dodou 2015 (and Masicampo & Lalande 2012) by focusing on the re-analysis and re-interpretatation of their data.

The text can be better structured with one subsection for each alternative model/explanation. I would like to see more rigour in the alternative explanations put forward. The mathematics/statistics behind each model should be fully explained. And, statistics has a means to compare data-based models. So why are there no AICs in the paper.

To put the discussion on p-values and p-value hacking into context, I would appreciate a citation to (Gelman & Stern 2006).

Details (for the sake of time, written quickly, and with the view of an outsider as the likely reader of the paper):

p.69: 0.01 -> 0.001?
p.76-83. Break up some of these sentences. They are hard to read.
p.86 four processes? You mention four things that from the outside are more than just parameters. Later you turn them into four parameters.
P89 the true Type I error here and number of true H0 and true H1 in line 87 are related, are they not? ‘ Inflated’ on line 90 now relates to mechanism, not the type I error itself. This is not what is intended, is it?
P111 ‘might’ and ‘ assume’ cannot go together.
p.125 hypothesis-> null hypothesis?
p.171 two -> three? the ratio of true to false effects is of influence also?
p.180 table below -> table 1
p.170 ‘we can expect’ How? Specify explicitly the simple model you use here. If too lengthy, make it supplementary.
p.194 The first column with 0.01s is not simple what the caption says.
p.197 I failed to see what ratios are calculated.
Table 3: N/T not explained. Please explain better what you do here.
p.222-223 “ is false’ missing?? Please recall that all models are wrong and some are useful.
p.239 how?
p.240 end: add: as we show now. (?)
p.248-251. This is not yet related to Table 4??
Table 4: the Type 1 error rate change 2013 is nothing very special, a monotonic increase. Is not very unlikely, is it?
p.310 Too strong a conclusion. Cannot be drawn from this study. They are not even addressed.
p.311-312 Delete “ Although” . How can this be an opposition, it is an and-and. And the statement is an ‘of course’ .

Gelman A, and Stern H. 2006. The Difference Between “Significant” and “Not Significant” is not Itself Statistically Significant. The American Statistician 60:328-331.

I look forward to your revision.

·

Basic reporting

*** Review of commentary article “A peculiar surge of incorrect conclusions about the prevalence of p-values just below .05”

The commentary includes several observations that are useful, interesting, and complementary to the findings by De Winter and Dodou (2015). However, the commentary also contains several ambiguous and contradictory statements. Furthermore, the author does not make clear how his hypothesized publication-bias mechanism works, how it relates to the scientific literature, and how it differs from other known biases that affect the p-value distribution. In addition, the author does not sufficiently discuss the validity of his p-curve model. Overall, the author brings up many diverse points, but fails to provide a clear conclusion.

*** Ambiguous overall message
Based on my reading, I am unclear which of the following possible messages the author wants to convey:
1. De Winter & Dodou (2015) make “incorrect conclusions”.
2. De Winter & Dodou (2015), Masicampo and Lalande (2012), and Leggett et al. (2013) claim there is a rise of p-values just below 0.05, but they do not provide “substantial evidence for such a rise”.
3. The “observed differences provide no indication for a surge of p-values” just below 0.05.
4. The “observed differences provide no indication for a surge of p-values … due to an increase in questionable research practices”.
5. It is “not very likely to be true that there is ‘a surge of p-values’” just below 0.05.
6. The data of De Winter and Dodou do not “suggest there is an increase in questionable research practices over the last 25 years”.
7. The data of De Winter and Dodou do not support a “surge of p-hacking”.
8. The data of De Winter and Dodou are “very unlikely to occur when p-hacking”.
9. The results by De Winter and Dodou (2015) are “difficult to explain based on the idea that questionable research practices have increased”.
10. The results of De Winter and Dodou (2015) are “not in line with an explanation based on an increase in questionable research practices over time”.
11. There have been (substantial) longitudinal changes in the p-value distribution, and they “can be explained by an increase in publication bias over the years”.
12. There have been (substantial) longitudinal changes in the p-value distribution, and it “seems likely” they have been caused by an increase of publication bias over the years.
13. There have been substantial longitudinal changes in the p-value distribution, but they are “due to publication bias increasing, instead of due to an increase in p-hacking”.
14. “Publication bias and underpowered studies are a much bigger problem for science than inflated Type 1 error rates” (note that this statement does not refer to a surge or rise at all, but to prevalence instead).
15. “It is time researchers investigating inflated Type 1 errors use better models, make better predictions, and collect better data”.

The above statements all mean something different. For example, there is a difference between saying something is incorrect, saying there is no substantial evidence for something, and saying that something is (un)likely. If a statement is intended as a refutation of De Winter & Dodou (2015), Masicampo & Lalande (2012), and/or Leggett et al. (2013), then this needs to be made more explicit in relation to the original statement made by the respective authors.

Note also that De Winter and Dodou (2015) did not state there is a “surge of p-hacking”, so it is not sensible to target such a claim. De Winter and Dodou (2015) concluded: “The fact that p-values just below 0.05 exhibited the fastest increase among all p-value ranges we searched for suggests (but does not prove) that questionable research practices have increased over the past 25 years. This interpretation should be regarded with some caution, since a change of the p-value distribution can occur for various reasons, including changes in the file drawer effect and changes in the number of studies investigating a true effect (see also Lakens, in press).” (Italics added). The author of the commentary does not make clear what exactly he tries to refute or has to add compared to the conclusions.

Several of the aforementioned statements are contradictory with statement 15. You cannot use a model on data of De Winter and Dodou (2015) to refute claims by De Winter and Dodou (2015), while at the same time conclude that better models need to be developed. It cannot be both. Does the author think his p-curve model (1) is valid, (2) can be valid, (3) or is invalid?

The author concludes that “publication bias and underpowered studies are a much bigger problem for science than inflated Type 1 error rates”, but he does not sufficiently relate this statement to prior evidence. There is a body of literature arguing the exact opposite. Ioannidis (2010), for example, claimed: “I suspect that publication bias for whole studies is not the most common problem in most scientific fields. Selective reporting biases affecting specific outcomes and specific analyses within studies is probably the greatest and most intangible concern that distorts the literature across many fields (Chan and Altman, 2005; Chan et al., 2004a,b; Contopoulos-Ioannidis et al., 2006; Mathieu et al., 2009)”. It is an over-generalization to conclude that “publication bias and underpowered studies are a much bigger problem for science than inflated Type 1 error rates” based on the fit of a single model. Furthermore, it is unclear how the author defines a “problem” from a cost-benefit (or detection theory) perspective.

The title “A peculiar surge of incorrect conclusions about the prevalence of p-values just below .05” is ambiguous. It is an amalgam of the title by Masicampo and Lalande (2012): “A peculiar prevalence of p values just below .05” and the title by De Winter and Dodou (2015): “A surge of p-values between 0.041 and 0.049 in recent decades (but negative results are increasing rapidly too)”. However, it is not clear whether the commentary targets the article by Masicampo and Lalande (2012) and/or the article by De Winter and Dodou (2015). The commentary also fails to make explicit what exactly are those “incorrect conclusions”. Note that there are important differences between the results of these two articles: Masicampo and Lalande (2012, see also Leggett et al., 2013) found a peak of p-values just below 0.05, while De Winter and Dodou (2015) stated “we did not find a peak of p-values just below 0.05”. De Winter and Dodou (2015) also cite, and agree with, Lakens’ (in press) previous commentary on the work by Masicampo and Lalande (2012).

*** Not making clear what the difference/similarity is between questionable research practices (QRPs), p-hacking, and publication bias
De Winter and Dodou (2015) found that among the three-digit p-values, the strongest 1990–2013 increase has occurred for p-values in the range 0.041–0.049, while the least strong increase has occurred for p-values in the range 0.051–0.059. Similar results were found for 2-digit p-values, with the strongest increase for p = 0.04 and the least strong increase for p = 0.05. One possible (intuitive and parsimonious) explanation for these trends is that questionable research practices (QRP) have increased over the years. De Winter and Dodou (2015) used QRP as an umbrella term for behaviors that turn a negative result into a positive result. This use of the term QRP is in line with John et al. (2012), who considered QRPs to be the “exploitation of the gray area of acceptable practice”. Ioannidis et al. (2014) considers publication bias to be a special case of QRP (“forms of publication and reporting biases and other questionable research practices”).

The commentary argues that the longitudinal trends in p-values observed by De Winter and Dodou (2015) are caused by a specific form of publication bias, whereby “p-values between .051-0.59 [sic] (or ‘marginally significant’ results) were more readily interpreted as support for the hypothesis in 1990-1997 than in 2005-2013”. The author argues that this type of publication bias is caused by a change of “leniency of reviewers and editors to accept marginally significant p-values as support for a hypothesis”. This novel publication-bias mechanism matches the QRP explanation proposed by De Winter and Dodou (2015). It is unclear why the commentary does not explicitly acknowledge the similarities between both types of ‘negative’ findings. Both mechanisms (i.e., accepting marginally significant p-values as support for a hypothesis vs. turning a negative result into a positive result due to selective analysis and reporting) seem statistically almost indistinguishable.

The author of the commentary cites Fanelli (2012) to support his point: “previous research has revealed there is an increase in publication bias over the years (Fanelli, 2012)”. However, Fanelli (2012) did not show this. In his paper, Fanelli (2012) showed that the percentage of positive results has increased with respect to all hypothesis-testing papers. Fanelli provides several potential explanations, including publication bias as well as QRPs: “Negative results could be submitted and accepted for publication less frequently, or somehow turned into positive results through post hoc re-interpretation, re-analysis, selection or various forms of manipulation/fabrication.” (p. 899). Thus, the author’s claim that Fanelli (2012) revealed an increase of publication bias reflects a very poor interpretation of the literature.

*** Concerns about model validity and contradictory conclusions
The author of the commentary fits a model in order to come up with conclusions about growing publication bias and a reduction of statistical power. However, no evidence is provided whether the proposed model is, or can be, valid.

De Winter and Dodou (2015) discussed various alternative explanations for changes in p-curve distributions. One such alternative explanation is that researchers have changed their ways of reporting, from textual expressions of statistical significance towards numeric p-values. Furthermore, the model in the commentary assumes homogenous effect sizes, while in reality, effect sizes should have a certain distribution. Curiously, the author concludes: “Analyzing huge numbers of p-values, which come from studies with large heterogeneity, will not be able to provide any indication of the prevalence of questionable research practices”. However, this is exactly what he is doing when he says: “the analyses in the present article point to the fact that low statistical power and publication bias, and not p-hacking, are the biggest problems in the scientific literature”.

References
De Winter, J. C. F., & Dodou, D. (2015). A surge of p-values between 0.041 and 0.049 in recent decades (but negative results are increasing rapidly too). PeerJ, 3, e733.
Fanelli, D. (2012). Negative results are disappearing from most disciplines and countries. Scientometrics, 90, 891–904.
Ioannidis, J. (2010). Meta‐research: The art of getting it wrong. Research Synthesis Methods, 1, 169–184.
Ioannidis, J. P., Munafò, M. R., Fusar-Poli, P., Nosek, B. A., & David, S. P. (2014). Publication and other reporting biases in cognitive sciences: detection, prevalence, and prevention. Trends in Cognitive Sciences, 18, 235–241.
John, L. K., Loewenstein, G., & Prelec, D. (2012). Measuring the prevalence of questionable research practices with incentives for truth telling. Psychological Science, 524–532.
Lakens, D. (in press). What p-hacking really looks like: A comment on Masicampo & Lalande (2012). Quarterly Journal of Experimental Psychology.
Leggett, N. C., Thomas, N. A., Loetscher, T., & Nicholls, M. E. (2013). The life of p: “Just significant” results are on the rise. The Quarterly Journal of Experimental Psychology, 66, 2303–2309.
Masicampo, E. J., & Lalande, D. R. (2012). A peculiar prevalence of p values just below. 05. The Quarterly Journal of Experimental Psychology, 65, 2271–2279.

Experimental design

-

Validity of the findings

-

Additional comments

-

·

Basic reporting

The paper makes an important contribution to a growing literature and debate on the prevalence and nature of resaerch and publication bias. It is well written and clear, and offers a constructive criticism to a paper previously published in PeerJ. The amount of background literature and detail provided are adequate to the scopes of the paper.

Cited papers seem to be missing from the reference list.

Experimental design

"No Comments"

Validity of the findings

The findings are valid, and are based on a sound theoretical argument, sufficiently prudent in its assumptions and conclusions.
I believe that their interpretation, however, could benefit from some additional specifications. In particular, the author uses the term "publication bias" in opposition to p-hacking, but they should avoid conflating all non-p-hacking behaviours into a one category. The author seems to use such term to indicate editorial and peer-review rejection biases as well as a variety of questionable research and reporting practices including: selective reporting of results from one study, complete non-publication of non-significant results (i.e. "real" file-drawer problem), biased reporting of results, or even just a "simple" over-emphasis of significant results in abstracts. the consequences and implications of these behaviours are very different from one another. The latter, in particular, would imply that contemporary researchers are increasingly omitting or reporting in words their non-significant results in the abstract whilst still or increasingly using exact P-values for their significant results, all the while in the full-text they keep reporting everything in a complete and unbiased fashion. If this were the case, then scientific knowledge would not be under a growing threat of being distorted, and instead would be simply undergoing changes in style and content of abstracts, a trend that may be just as interesting and perhaps just as questionable as that of a growing file-drawer problem, but certainly not as worrying.

The author might be not interested in examining each behavioural scenario in the text, but he should at least point out that by "publication bias" he means all the above behaviours and more, and/or spend a sentence or two to distinguish between the potential effects of each practice, and their potential implications for the future of science. Fanelli 2012 but also Fanelli 2010 (PLoS ONE - DOI:10.1371/journal.pone.0010068) briefly discussed the different possibilities (but note that it only considered the possibility of a GROWING, not a DECREASING statistical power. The latter is a very important new insight into the complexity of the phenomena in question).

Finally, I think Pautasso 2010 (DOI 10.1007/s11192-010-0233-5) should get credit for first suggesting a growth in file-drawer effects "in abstracts", as he correctly put it, which is what de Winter and Dodou (but not Fanelli 2012) are actually measuring.

---

## Round 0.2 · Minor Revisions

· Academic Editor

Minor Revisions

The clarity of exposition has much improved except for the L259-280 (Changes in Type 1 error rates over time) that remain unclear to me. Version 0 has near current line 166 “it turns out that”, so that I would suggest to add to line 267 “as we show now”. But when at line 279 I still have the questions: how can it be reproduced and at 280 why it is unlikely. This must be clarified in the paper before I can accept the ms. The manuscript further requires more careful proofreading as I found the many typos (see below) and errors in the literature list.

Details

Table 3 caption. I suggest to change ratio to fraction and Ratio to ratio of fractions. Adapt the main text correspondingly. Now ratio in text can mean two things. Suggested caption “Reconstructed and observed fractions of p-values in bins between 0.001 and 0.049 and 1990/1913 ratios thereof. The observed values are from De Winter and Dodou (2015). N= number of p-values in bins, T = total number of p-values.”

L214 after .300 insert: giving 1684 and, without rounding in Table 1, 1676 in Table 2

Literature list misses Fanelli 2012 and Pautasso 2010 and one Lakens is misplaced.

Typos found:
L95 processes -> factors (as in line 182).
L123 en up -> end up
L219 but the -> by the
L233 a model -> the model
L234 move Similarly to after ‘differences’
L265 closing parathesis without initial one.
L267 in not -> is not
L345 require diff -> required diff
L384 Coda -> Conclusions
L300-301. Replace the numbers by (see Table 4, column 2). And delete its mention on L300.

---

## Round 0.3 · Minor Revisions

· Academic Editor

Minor Revisions

Thank you for the constructive response. I respond somewhat later, as I was out of office.I have one probably minor point left. On line 272 I did not understand the 0.08%. I expected 5%/5bins = 1% or perhaps 4.98/5. Please comment and revise.

---

## Round 0.4 · accepted · Accept

· Academic Editor

Accept

Thank you for the update on this last point. The paper is now clear to me and, I expect, many non-insiders to the issue. I hope many can learn from the approach proposed in the paper.